# Carotenoid Content and Profiles of Pumpkin Products and By-Products

**DOI:** 10.3390/molecules28020858

**Published:** 2023-01-15

**Authors:** Antonela Ninčević Grassino, Suzana Rimac Brnčić, Marija Badanjak Sabolović, Jana Šic Žlabur, Roko Marović, Mladen Brnčić

**Affiliations:** 1Faculty of Food Technology and Biotechnology, University of Zagreb, Pierottiejva 6, 10000 Zagreb, Croatia; 2Faculty of Agriculture, University of Zagreb, Svetošimunska cesta 25, 10000 Zagreb, Croatia

**Keywords:** carotenoids, seed, peel, pulp, individual carotenoids, functional value

## Abstract

The goal of this review is to provide an overview of the current findings on the major carotenoids and their content in pumpkin products and by-products. The content of total carotenoids and the composition of carotenoids in pumpkins depend mainly on the species and cultivar, pedoclimatic conditions, the part of the plant (pulp, peel or seed), extraction procedures and the type of solvent used for extraction. The major carotenoids identified in pumpkins were β-carotene, α-carotene, lutein and zeaxanthin. β-Carotene is the major carotenoid in most pumpkin species. The number and content of total carotenoids are higher when minor carotenoids and ester forms are considered. The use of carotenoids in the development of functional foods has been the topic of many versatile studies in recent years, as they add significant value to foods associated with numerous health benefits. In view of this, pumpkin and pumpkin by-products can serve as a valuable source of carotenoids.

## 1. Introduction

The pumpkin (*Cucurbita* L.) is a squash fruit vegetable belonging to the Cucurbitaceae family, also known as the gourd family, with 130 genera and 800 species. *Cucurbita* is native to Latin America and has been cultivated in Europe for over 500 years. The genus *Cucurbita* includes five domesticated species (*Cucurbita argyrosperma*, *Cucurbita ficifolia*, *Cucurbita maxima*, *Cucurbita moschata* and *Cucurbita pepo*), of which *Cucurbita moschata*, *Cucurbita maxima* and *Cucurbita pepo* are the most economically important species cultivated worldwide [1] and are widely used in the food industry for the commercial production of pumpkin pie, flour, seed oil, seeds as snacks, bread, cookies, desserts, cereals, ice cream, pancakes, puddings, pumpkin butter, salads, soups and stuffings [2]. In addition to the different pumpkin species, there are also numerous varieties that differ from the same species in chemical composition, color, shape and, due to agroclimatic conditions, agrotechnical measures. However, with some differences between varieties, all parts of the pumpkin plant, i.e., fruits, flowers, leaves, roots, stems and seeds, are edible.

Currently, the world production of pumpkin (including squash and gourds) amounts to approximately 27 million tons, of which China is the main producer with 5.5 million tons. In Europe, the current production of pumpkin for consumption is more than 4 million tons [3]. The huge global production of pumpkin and its consumption, e.g., in cooked, baked, and processed forms, is probably related to the growing interest of consumers in consuming a wide range of nutrients and phytochemicals through an adequate and balanced diet. As a result of this great use of pumpkin, its by-products are produced. In addition, it is important to emphasize that the cultivation and processing of pumpkin seeds produce by-products in the form of peel and pulp. On the other hand, when pumpkin is used for cooking, the peel and seeds are by-products. A good example of the problem of pumpkin by-products is the production of pumpkin seeds. For example, the production of pumpkin seeds from *C. maxima* produces about 49 times the amount of by-products (pulp and peel) than the number of seeds produced [4]. In general, the processing of pumpkin produces about 72–76% pulp, 2.6–16% peels and 3.1–4.4% seeds [5].

The by-product fractions are underutilized and are usually used for animal feed enrichment [6], but due to the phytochemical densities, these fractions have economic potential and could be explored for various other applications, including those with therapeutic and pharmacological benefits for human health, such as antibacterial, antiparasitic, antioxidant and pro-oxidant, carcinopreventive, antidiabetic, analgesic and anti-inflammatory [7,8,9,10,11,12].

Pumpkin by-products could also be used as a source of valuable components in cosmetic and food industries for the fortification of various products, as biodegradable food packaging and as carriers in encapsulation processes [2,13,14,15,16,17,18]. From a nutritional point of view, pumpkin fruits, i.e., pulp, peel and seeds, contain carbohydrates, proteins, lipids, fiber and minerals [19,20]. For example, partially defatted pumpkin seed flour obtained from *Cucurbita maxima*, which is a by-product of pumpkin seed oil production, is considered a good source of proteins (42.75%), lipids (12.28%) and total carbohydrates (37.4%), of which 26.64% is crude fiber [21]. This partially defatted pumpkin seed flour also contains high amounts of minerals such as potassium (1290 mg/100 g), magnesium (693 mg/100 g), iron (87.8 mg/100 g), zinc (11.5 mg/100 g) and copper (2.49 mg/100 g). Pumpkin fruits are also rich in phenolic compounds, fatty acids, essential amino acids, vitamins, terpenoids, saponins, sterols, tocopherols and carotenoids [9,19,22,23,24]. Their content in pumpkin seeds is highly variable and depends mainly on the pumpkin species and varieties, as well as on the fractions excreted.

Nowadays, by-products of cucurbits have also become very attractive as health-promoting ingredients and natural pigments with multiple uses due to the presence of carotenoids [25]. In the human body, carotenoids maintain chemical reactivity by scavenging free radicals and active atomic oxygen, which is particularly important for hard-working people. Carotenoids provide significant added value to foods associated with antioxidant and anti-cancer activity, photoprotection, cardiovascular disease protection and anti-inflammatory effects. Therefore, the use of carotenoids in the development of nutritionally enhanced functional foods and nutraceuticals from waste and by-products is one of the major challenges for many researchers today. In this context, the present work describes the potential of pumpkins, their waste fractions and pumpkin products to be used as a source of carotenoids. Since carotenoids have an unsaturated chemical structure and are, therefore, susceptible to reactions such as oxidation and isomerization, it is necessary to consider their stability during the processing and storage of pumpkin products. Here, the special focus is on the content and carotenoid profile of pumpkin products and pumpkin by-products.

## 2. Carotenoids: Sources, Diversity and Chemical Structure

Carotenoids are lipid-soluble natural pigments of the tetraterpenoid group that can be biosynthesized in plants, yeasts, fungi, algae and photosynthetic bacteria. They are responsible for the yellow, orange and red colors of various fruits and vegetables as well as processed foods of plant origin. Carotenoids consist of a 40-carbon skeleton of isoprene units covalently linked together, resulting in multiple conjugated double bonds. The structure may be cyclized at one or both ends with various degrees of hydrogenation and has oxygen-containing functional groups. Lycopene and α-carotene, for example, are acyclized and cyclized carotenoids, respectively. The most characteristic feature of carotenoids is the long series of conjugated double bonds that form the central part of the molecule. This gives them their shape, chemical reactivity and light-absorbing properties. Most naturally occurring carotenoids have the *trans* configuration, but they are isomerized into the *cis* configuration by conjugation during processing or by exposure to certain environmental conditions (light and heat). The isomeric form of carotenoids affects their functions in biological tissues, such as bioavailability, vitamin A activity, stability to electrophiles and specificity to cleaving enzymes [26].

Carotenoids are classified into two main groups: carotenes and xanthophylls. The carotenes are hydrocarbons (they consist only of carbon and hydrogen) and the xanthophylls contain oxygen as a functional group in their structure. Although more than 700 natural carotenoids have been identified, α- and β-carotene, lycopene, lutein, zeaxanthin and β-cryptoxanthin (PubChem) are the most widely distributed (Table 1).

Pumpkin contains large amounts of pro-vitamin A carotenoids, which give it different colors due to the presence of components such as lutein (bright-yellow color) and β-carotene (orange color). β-Carotene contains 100% of pro-vitamin A activity and, after conversion to vitamin A in the body, is responsible for night vision, growth and the development and maintenance of epithelial tissue and also has an immunological function. Although lycopene is one of the most studied carotenoids, it is present in pumpkins only in low concentrations. In addition, xanthophylls such as lutein, zeaxanthin and cryptoxanthin, as well as carotenes (α-carotene, β-carotene and lycopene), have been detected by many researchers in varying concentrations, depending mainly on the pumpkin variety and the way the pumpkins are treated [27].

### 2.1. Carotenoids in Cucurbita spp. Varieties

Numerous scientific papers have demonstrated a great diversity of carotenoids in different pumpkin varieties. Both the total content of carotenoids (Table 2) and the content of individual carotenoids (Table 3) are highly variable and depend on the variety and species (genetic characteristics), pedoclimatic conditions and agrotechnical measures in the cultivation of the pumpkin, the part of the plant (pulp, peel or seed) and extraction procedures (technique, type of solvent, mass ratio, etc.). For example, in the work of de Carvalho [28], it was shown that the mean total carotenoid content in two landrace samples of raw pumpkin pulp (*C. moschata*) is different, being 404.98 μg/g in sample A and 234.21 μg/g in sample B, respectively. Hussain et al. [29] investigated the total carotenoid content in three fractions of pumpkin (peel, pulp, and seed powder). The highest content of total carotenoids was found in pumpkin pulp powder (35.2 mg/100 g), while the lowest content of total carotenoids was found in pumpkin seed powder (8.2 mg/100 g). According to Kreck et al. [30], the total carotenoid content of pumpkin peel and pulp depends on the variety. Their results showed that the cultivar ‘Butternut’ had lower concentrations of total carotenoids in pulp (44 mg/kg) and peel (12 mg/kg) than the cultivar ‘Hokkaido’ pulp (218 mg/kg) and peel (1048 mg/kg). Total carotenoids in *C. maxima* seed oil were 107.5 and 32.5 μg/kg, at 440 and 460 nm, respectively [31]. In the peel and pulp of *C. maxima*, *C. moschata* and *C. andreana*, the total carotenoid content ranged from 0.2 to 19.57 mg/100 g, depending on the number of varieties of the three *Cucurbita* spp. evaluated [4]. In the lipid fraction of pumpkin seeds of ‘Nova caravela’, ‘Mini Paulisa’, ‘Menina Brasileira’ (*C. moschata*) and ‘Moranga de Mesa’ (*C. maxima*) cultivars, the most appreciated by consumers, the values of total carotenoids ranged from 7.67 to 26.80 µg/g [32]. In the oils obtained by the cold pressing of *C. pepo* seeds, the amounts of total carotenoids ranged from 6.95 to 7.60 μmol/kg depending on the growing region [33]. The total carotenoid content, expressed as β-carotene, in oils of *C. maxima*, ‘Moranga de Mesa’ variety, obtained from a mixture of peels and seeds by ultrasound-assisted extraction (UAE) and Soxhlet extraction, is high, ranging from 225.43 to 296.94 mg/100 g of oil, depending on the method and temperature [34]. Compared to the oil obtained from the mixture of peels and seeds, the seed oil extracted with UAE at 75 °C contained an extremely low mass fraction of β-carotene (1.5 mg per 100 g of oil). In another study, the effects of different extraction conditions of carotenoids from pumpkin pulp were investigated. Carotenoid extraction was optimized for temperature (15, 30 and 45 °C), extraction time (8, 12 and 16 h), and solid-to-solvent ratio (1:50, 1:100 and 1:150). The authors observed that nonpolar solvents were a critical factor for the extraction of nonpolar carotenoids such as lycopene and β-carotene from pumpkin pulp [35]. It was also found that the yield of carotenoids was increased when the ratio of solid to solvent was increased up to 1:150. In another study, Portillo-Lopez et al. [36] used canola, corn and soybean oils at different solid-to-solvent ratios (1:10, 2:10 and 3:10) as alternative green solvents for the extraction of carotenoids from pumpkin pulp (*C. argyrosperma*). Before extraction, the pumpkin pulp slices were dried at 60 °C, ground and sieved (0.425 mm) to obtain a fine powder. The obtained results showed that the type of solvent and the solid–liquid ratio significantly affected the carotenoid yield. The highest carotenoid content was obtained with canola oil due to its physical (e.g., boiling point, surface tension and viscosity) and molecular properties (e.g., dipole moment, electronic polarizability, ability to release and accept hydrogen bonds and ability to release and accept electron pairs). It was also found that the solid-to-solvent ratio of 1:10 g/mL gave the highest yield of carotenoids. Sharma and Bhat [37] used corn oil as a green solvent and three different extraction techniques (ultrasound-assisted extraction, microwave-assisted extraction and conventional extraction) for carotenoid extraction from the pulp and peel of two pumpkin (*C. maxima*) cultivars. A higher carotenoid content was found in the samples after ultrasound-assisted extraction than in the samples after microwave-assisted extraction and conventional extraction with organic solvents (hexane and isopropyl alcohol). In addition, it was found that the carotenoid content varied depending on the plant part, and the peel powder had a higher content of carotenoids than the pulp powder. Several studies have also reported the recovery of carotenoids (lycopene and β-carotene) from food waste and by-products (pomegranate waste and mango pulp) using vegetable oils as solvents [38,39].

### 2.2. Overview of the Individual Carotenoid Characteristics for Cucurbita spp. Varieties

In addition to the determination of the total carotenoid, the results of the analyses of the individual carotenoids are also described in the present work (Table 3). The composition of carotenoids in pumpkins can be influenced by some factors such as differences in ripening stage, pedoclimatic conditions and harvest and postharvest treatment. Fruit ripening is generally associated with increased carotenogenesis and, consequently, an increased concentration of carotenoids during ripening. Normally, the biosynthesis of carotenoids decreases at lower temperatures. Four carotenoids are predominant in pumpkins. β-Carotene is the major carotenoid in most species. The number of total carotenoids may be higher when minor carotenoids and ester forms are considered. Other carotenoids such as α-carotene, lutein, and zeaxanthin are also abundant. Procida et al. [51] showed that commercial pumpkin seed oil from *C. pepo* contains 93–121 mg/kg lutein and 98–116 mg/kg zeaxanthin. Although the total amount of lutein and zeaxanthin is very similar, the availability of the two carotenoids seems to be different. It was found that after hydrolysis of the oil to obtain the free, esterified and total content of lutein and zeaxanthin, the ratio between the free and esterified fractions is significantly different. For example, zeaxanthin is present mainly in esterified form in all samples, while lutein is found in esterified form in only three samples. A study by Akin et al. [33] revealed that *C. pepo* cold-pressed oil contains low quantities of lutein, β-carotene, β-cryptoxanthin and zeaxanthin. Using high-performance liquid chromatography analysis with diode array detector, electrospray ionization and tandem mass spectrometry (HPLC-DAD-ESI/MS/MS), 0.02–0.03 mg/100 g lutein, 0.54–0.6 mg/100 g β-carotene, 0.43–0.49 mg/100 g cryptoxanthin and 2.65–2.91 mg/100 g zeaxanthin were found.

Other studies indicated that oils extracted from seeds can also be used as an excellent source of carotenoids. For example, seed oil extracted from *C. maxima*, variety ‘Berrettina’, dried at 60 °C for 24 h in a hot air oven and analyzed by HPLC with diode array and mass spectrometry detection (HPLC-DAD-MS) contained two important carotenoids, namely, lutein and β-carotene at concentrations of 8 mg/L and 2.5 mg/L, respectively [52]. The oil obtained by the cold pressing and aqueous enzymatic extraction of *C. pepo* seeds also contained lutein and β-carotene [53]. Their mass fraction varied between 30.1 and 38.7% and 25.3 and 28.1% for lutein and β-carotene, respectively, depending on the extraction methods used. The (9*Z*)-β-carotene, α-carotene and traces of zeaxanthin were also identified. The carotenoids were also analyzed in roasted and unroasted seeds, and it was found that the roasting method had a great influence on the carotenoid content, i.e., the total carotenoid content increased from 0.29 to 0.54 mg/100 g, from unroasted to roasted [55]. The major carotenoid is lutein, which accounted for about 44% of the total carotenoid content and increased to 50% after roasting. β-Carotene is the second most abundant carotenoid, and its content in unroasted and roasted pumpkin seeds is 43 and 39%, respectively. The content of cryptoxanthin in unroasted seeds is twice as high as in roasted seeds.

In addition to seeds, the other pumpkin fractions, such as peel and pulp, also contained high amounts of carotenoids. Thus, β-carotene and β-cryptoxanthin are found in the peel, seeds and pulp of *C. pepo*, *C. moschata* and *C. maxima* [54]. Measurement by HPLC with UV/VIS detection showed that the peels contained more β-carotene than the other parts. The values ranged from 39.48 to 123.19 mg/kg, depending on the species. The seed fraction of *C. maxima* also contained a high amount of β-carotene (31.40 mg/kg). β-Cryptoxanthin was detected only in the pulp of *C. maxima* (0.65 mg/kg), in the peels of *C. pepo*, *C. moschata* and *C. maxima* (0.13–6.52 mg/kg), and in the seeds of *C. maxima* (0.21 mg/kg) and *C. pepo* (0.16 mg/kg). In the work of de Carvalho et al. [28], HPLC analysis was used to determine α- and β-carotenes and their isomers, (9*Z*)- and (13*Z*)-β-carotene, in the pulp of two landrace samples (A and B) of raw *C. moschata* to verify its production potential. The results showed that (all-*E*)-β-carotene was the most abundant in both landrace materials, containing 244.22 and 141.95 μg/g in samples A and B, respectively. α-carotene is found in amounts ranging from 67.06 μg/g (sample A) to 72.99 μg/g (sample B). The levels of (9*Z*)-β-carotene and (13*Z*)-β-carotene were relatively negligible compared to the levels of α- and β-carotene in landrace samples A and B, respectively. The values for (9*Z*)-β-carotene were 2.34 μg/g (A) and 0.97 μg/g (B), and those for (13*Z*)-β-carotene were 3.67 (A) to 1.84 μg/g (B). In addition to the various plant parts, carotenoids were also determined in various pumpkin products such as pumpkin juice, pumpkin puree and similar products. Suo et al. [56] investigated the carotenoid content in pumpkin juice and the effect of sonication of pumpkin juice on carotenoid content. The results showed that sonication led to a significant increase in the content of carotenoids in the sonicated juice (22.83–32.28% in β-carotene, 23.57–32.52% in α-carotene, 22.55–32.33% in β-cryptoxanthin, 23.70–32.59% in zeaxanthin and 24.14–34.48% in lycopene). Provesi et al. [57] investigated by HPLC the concentration of the main carotenoids in the pumpkin puree of two pumpkin cultivars, *C. moschata* ‘Menina Brasileira’ and *C. maxima* ‘Exposição’ and reported that the most abundant carotenoids in the pumpkin puree of *C. moschata* ‘Menina Brasileira’ were (all-*E*)-β-carotene and α-carotene, with lesser amounts of ψ-carotene, violaxanthin and lutein. The most abundant carotenoid in the samples of *C. maxima* ‘Exposição’ was (all-*E*)-β-carotene, followed by violaxanthin and lutein. The retentions of carotenes after heat treatment, namely, α-carotene and (all-*E*)-β-carotene in *C. moschata* ‘Menina Brasileira’ squashes and (all-*E*)-β-carotene in *C. maxima* ‘Exposição’ squashes, were high (>75%). Norshazila et al. [35] reported a significantly higher total carotenoid content at 30 °C than at 45 °C. According to Oliveira et al. [60], the degradation of carotenoids occurs at temperatures close to 40 °C. Moreover, Das and Bera [61] found that the maximum β-carotene extraction yield was at 40 °C and decreased with a further increase in temperature. Kurz et al. [62] noted the presence of relatively high levels of β-carotene and lutein in eight investigated pumpkin varieties of three pumpkin species (*Cucurbita maxima* Duch., *Cucurbita pepo* L. and *Cucurbita moschata* Duch.)

Carotenoids were also determined in pumpkin-enriched corn grits extrudates [58]. The peeled, oven-dried at 60 °C, ground and powdered pumpkins with mass fractions of 4, 6 and 8% were added to corn grits with and without the addition of ascorbic acid. The results showed that the content of lutein and zeaxanthin in the extruded samples is higher than in the raw material. The amount of lutein ranges from 3.16 to 7.93 mg/100 g, depending on whether ascorbic acid and pumpkin powder were added to the raw material or not. After extrusion, the amounts of lutein range from 4.79–12.29 mg/100 g to 3.62–10.89 mg/100 g, depending on the temperatures used. Zeaxanthin was found in amounts of 1.74–8.93 mg/100 g in the raw material and 2.94–10.53 and 2.83–9.39 mg/100 g in the extruded samples. α-Carotene showed low amounts at 0.03–0.18 mg/100 g, and the lowest values were obtained in the extruded samples, indicating that a high temperature has a decreasing effect in the case of this carotenoid. Moreover, (9*Z*)-β-carotene and (13*Z*)-β-carotene were found in low amounts, i.e., 0.03–0.33 mg/100 g and 0.023–0.14 mg/100 g, respectively. In recent years, edible flowers have become a culinary trend. Edible flowers that are now increasingly available on the market are squash blossoms. Biezanowska-Kopec et al. [50] studied the nutrient composition and antioxidant activity of squash flowers from different squash species (*C. maxima*, *C. moschata* and *C. pepo*) and cultivars ‘Amazonka’, ‘Ambar’, ‘Atlantic Giant’, ‘Bambino’, ‘Butternut’, ‘Muscade de Provence’, ‘Rouge vif d’Etampes’ and ‘Miranda’. The highest carotenoid content (45.82 mg/100 g) was found in the ‘Miranda’ cultivar (*C. pepo*), while the lowest content (15.27 mg/100 g) was found in the ‘Butternut’ cultivar (*C. moschata*). In addition to the pumpkin flowers, the pumpkin leaves can also be used for cooking. Mashiani et al. [59] studied the carotenoid profiles of African pumpkin (*Momordica balsamina* L.) leaves subjected to different cooking techniques. The results showed that the different cooking techniques increased the content of lutein, carotene and zeaxanthin in the pumpkin leaves.

## 3. Conclusions

The most studied Cucurbita species are *Cucurbita pepo, C. moschata* and *C. Maxima*. Pumpkin processing generated large amounts of by-products. Pumpkin by-products are still underutilized. The available research reports have pointed out how four carotenoids are predominant in pumpkins. β-Carotene is the major carotenoid in most species. Pumpkin seeds are mainly used for the production of pumpkin seed oil, which is used in cooking, but recently, it has also been used in the pharmaceutical and cosmetic industries. Lutein and zeaxanthin are the most abundant carotenoids in seed oil, while β-carotene is the most abundant carotenoid in pumpkin peel and pulp. The recovery of carotenoids from pumpkin by-products through sustainable green technologies could be used in the development of functional food products and pharmaceuticals to prevent chronic diseases and improve health. While the demand for carotenoids is increasing in the global carotenoid market, the amount of natural carotenoids extracted from plants, animals and microorganisms is very small. Future trends are aimed at better understanding for more efficient and environmentally friendly extraction methods of carotenoids that comply with green extraction principles and utilize non-conventional sustainable technologies such as supercritical fluid extraction and extraction using ultrasound, pulsed electric fields, microwaves and enzymatic treatments. In addition, the use of environmentally friendly bio-based solvents as a green alternative to petroleum-derived solvents such as hexane is a key requirement for sustainable development. In addition, hybrid technologies that merge different energy sources will also be a novelty in this field. These new findings will provide new opportunities for large-scale production of new natural ingredients and products in the food, feed, cosmetic and pharmaceutical industries based on pumpkin by-products.

## Figures and Tables

**Table 1 molecules-28-00858-t001:** Chemical structure of carotenoids found in *Cucurbita* spp.

Compound Trivial Name	Semisystematic Name	Empirical Formula	Structure
α-carotene	(6′R)-β,ε-carotene	C_40_H_56_	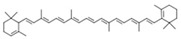
β-carotene	β,β-Carotene	C_40_H_56_	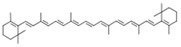
Lutein	(3R,3′R,6′R)-β,ε-carotene-3,3′-diol	C_40_H_56_O_2_	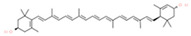
Zeaxanthin	(3R,3′R)-β,β-carotene-3,3′-diol	C_40_H_56_O_2_	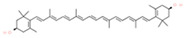
β-Cryptoxanthin	(3R)-β,β-caroten-3-ol	C_40_H_56_O	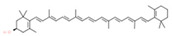

**Table 2 molecules-28-00858-t002:** Total content of carotenoids in pumpkin products and by-products.

Cucurbita Species	Variety	Products/By-Products	Extraction Solvent	Total Carotenoid Content (µg/g)	Reference
*C. moschata*	Duches A	Pulp	Acetone/ petroleum ether	404.98	[28]
	Duches B			234.21	
*C. maxima*	-	Seed oil	*n*-Hexane	0.107	[31]
*C. maxima*	Db1#	Peel and pulp	*n*-Hexane/ Acetone/ Ethanol	195.7	[4]
*C. moschata*	Nova Caravela	Lipid fraction from seeds		7.67	[32]
	Mini Paulista			26.80	
	Menina Brasileira		Chloroform/ Methanol	26.03	
*C. maxima*	Moranga de Mesa		Water	11.53	
*C. pepo*	-	Seed oil	-	69.5–76.0	[33]
*C. maxima*	Rouge	Pulp Peel	Acetone/hexane	683 1751	[30]
*C. moschata*	-	Pulp	Virgin coconut oil	171.96	[35]
*C. moschata*	Duch. cv. Miben	Peel	Ethanol/petroleum ether	344.9	[40]
*C. moschata*		Pulp	Acetone/Ethanol	367	[41]
*C. maxima*	-	Pulp	Petroleum ether	21.20	[42]
*C. maxima*	-	Oil from seed/peel mixture	Ethanol	2254–2960	[34]
		Seed oil		15	
*C. maxima*	-	Pulp	Acetone	675.62	[43]
*C. moschata*	Butternut Waltham Muscade de Provence Llena de Napoles Butternut Rugosa	Pulp	Acetone	143.8 3797.2	[44]
				881.8	
				338.7	
*C. moschata*	-	Pulp	Hexane/Acetone/Ethanol	10.09	[45]
*C. moschata*	Pluto	Pulp	Acetone	34.54	[46]
*C. maxima*	-	Peel powder Pulp powder Seed powder	Acetone/*n*-Hexane	237 352 82	[29]
*C. moschata*	-	Pulp	Acetone	196,7	[47]
*C. maxima*	-	Pulp	Acetone	41.1	[48]
*C. maxima*	Justynka	Pulp	Methanol/Ethyl acetate/light petroleum	487.4	[49]
*C. argyrosperma*		Pulp powder	Canola oil	613.9	[36]
*C. maxima*	Gold Nugget Amoro F1	Peel powder Pulp powder Peel powder Pulp powder	Hexane/Isopropanol	192.1 150.1 162.1 123.3	[37]
*C. maxima*	Amazonka Ambar Atlantic Giant Bambino	Pumpkin flower	Acetone/ *n*-Hexane	401.7 273.8 309.5 390.1	[50]
*C pepo*	Miranda	Pumpkin flower	Acetone/ *n*-Hexane	458.2	[50]
*C. moschata*	Muscade de Provence Butternut squash Rouge vif d’Etampes	Pumpkin flower	Acetone/ *n*-Hexane	213.6 152.7 184.4	[50]

**Table 3 molecules-28-00858-t003:** Content of major carotenoids in pumpkin products and by-products.

Cucurbita Species	Variety	Products /By-Products	Carotenoid/(µg/g)				Reference
			β-Carotene	α-Carotene	Lutein	Zeaxanthin	
*-*	-	Seed oil for dietary use Seed oil for therapeutic use	-	-	93–121 * 102–106 *	98–116 *	[51]
						109–113 *	
*C. pepo*	-	Seed oil	5.4–6.0 *	-	0.2–0.3 *	26.5–29.2	[33]
*C. moschata*	Ariel Pluto	Pulp	2.04 3.67	-	-	-	[46]
*C. maxima*	Berrettina	Seed oil	2.5 *		8 *		[52]
*C. pepo*	Herakles	Seed oil		17–23	63–88		[53]
	-	Pulp Peel Seed	1.48 39.48 17.46	-	-		
*C. pepo*						-	[54]

		Pulp	5.70	-	-		
*C. moschata*		Peel	68.30	-	-	-	[54]
		Seed	7.15	-	-		
		Pulp	17.04	-	-		
*C. maxima*		Peel	123.19	-	-	-	[54]
		Seed	31.40	-	-		
*C. pepo*	Oleifera	Seed oil	1.2–2.1	0.1–0.2	1.2–2.7		[55]
*C. moschata*	-	Juice	1.27	1.23	1.35		[56]
*C. moschata*	Menina Brasileira	Puree	19.45	12.60	0.59		[57]
*C. maxima*	Exposiçăo	Puree	13.38	0.43	10.43	-	[57]
*C. maxima*	Hokkaido	Pumpkin-enriched grits extrudates	2.1	0.03–0.18	36.2–108.9		[58]
*-*	*Momordica balsamina* L.	Pumpkin leaves	90	-	341.3	13	[59]

* (µg/mL).

## Data Availability

Not applicable.

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
