# Peer review of "Carotenoid Content and Profiles of Pumpkin Products and By-Products"

_molecules, 2023, doi:10.3390/molecules28020858_

Round 1

Reviewer 2 Report

Pumpkins and their derivatives are well recognized for their high carotenoid content. Numerous works throughout decades have described the contents and composition in different species, parts of the fruit, effects of agroclimatic conditions, etc. Although there are more general recent reviews focused on pumpkin bioactives or phytochemicals, or from a nutritional perspective and effects on health, there are no recent reviews on the profile of carotenoids and contents in pumpkin products and by-products. Due to the interest of carotenoids and their benefits for human health, I consider interesting to have an updated review that describes in detail the content and variability of carotenoids in this group of fruits and by-products.  Overall, this review compiles update information but needs to be improved. Some comments or suggestions are given below:

1.Table 1: I am not agree in using the term `synonyms´  for the different isomers of a carotenoid. It is well known that isomers of a carotenoid are not `equivalents ´in many aspects. For example, a particular cis isomer can be biosynthesized or catabolized by a specific pathway, and the accumulation, bioavailability and bioactivity of the different isomers for particular carotenoids can be very diverse. Therefore, I recommend to remove the `synonym´ column in the Table 1 and indicate the isomers that have been specifically detected in Cucurbita sp. The images of the structures are cropped.

The a- carotene and b-cryptoxanthin are also provitamin A carotenoids and should be included in the Table 1. 

2. In the description of carotenoids contents (total or individual) in the text (sections 2.1 and 2.2)  different units are use: migrog/g, mg/kg, mg/100 g, mg/L ; but in the Table 2 and Table 3 data have been transformed to microg/g (with some exceptions in Table 3 which are in microg/mL). The use of different units in the text makes very difficult to compare between data obtained in the different works. Again, the use of different units in text and tables leads to confusion for data interpretation and comparison. When possible, can authors use the same units?   

For not liquid products, are data based on fresh weight or dry weight? It is no indicated in the text.  

3. Can the data reported in sections 2.1 and 2.2 be organized according to criteria?  For example, first describing data from different varieties, species of fresh products (seed,  pulp, peel…), and then  data from processed or derived products (oil, puree…).   

Minor points:

4.Please, remark in the abstract which is in your opinion the novelty of this review. I think that the word `paper´ in line can be deleted.

5.Review or rewrite sentence in line 51-52. I cannot understand the meaning of: `` and one ton of seed squash can generate about 49  times more by-products than the amount of seeds´´ .

6.Please, write correctly `hexane´ in table 2

7. Review throughout the manuscript the symbol that has been used to indicate decimals in the carotenoid data. In table 2  both `dot´ and `comma´ are used for decimals.

8.Line 67. Write in italics the scientific name.

9.Some information seem to be duplicated . Please revise information in  lines 129-132, 190-192, 194-195.

Reviewer 3 Report

This study is about the carotenoid profile and their content in pumpkin products and by-products. It is an interesting study since pumpkin and its by-products are good carotenoid sources.

The abstract section well summarizes this important review article.

I also find the introduction part attractive and informative.

Please correct n-hexan as n-hexane throughout the MS.

All parts of this MS seem excellent except Conclusion. I think the authors did not pay attention to this part so much. I recommend writing a better conclusion particularly mentioning about the future trends in this area with some suggestions. 

I appreciate the authors for this work.

Round 2

Reviewer 1 Report

The authors took my comments into account and corrected the errors.

Author Response

Reviewer 1

The authors took my comments into account and corrected the errors.

AUTHORS ANSWER

We are very grateful for the helpful reviews and appreciate the encouraging comments provided by the reviewer of this manuscript. We are pleased that the reviewer is satisfied with the author's answers to the questions raised in his initial review.

Reviewer 2 Report

The authors have given satisfactory explanations to the previous comments.
Only one minor point remains to be clarified: The authors indicate that to facilitate the comparison, as it was suggested,
they have recalculated the units of concentration of the data shown in the tables and
the units indicate are mg/100g. However, in the body of the tables the units
are microg/g. Please, indicate which are the correct units in the tables or correct if necessary.

Author Response

Reviewer 2

The authors have given satisfactory explanations to the previous comments.

Only one minor point remains to be clarified: The authors indicate that to facilitate the comparison, as it was suggested, they have recalculated the units of concentration of the data shown in the tables and the units indicate are mg/100g. However, in the body of the tables the units are microg/g. Please, indicate which are the correct units in the tables or correct if necessary.  

AUTHORS ANSWER

We are very grateful for the comments and suggestions made by the reviewer of this manuscript. We have clarified how the units were presented in some articles: mg/100g; mg/kg, µg/g. We have recalculated all values in µg/g. Thus, the correct units in the tables are µg/g and no further correction is needed.